# Patterns of mental health problems before and after easing COVID-19 restrictions: Evidence from a 105248-subject survey in general population in China

**Depeng Jiang** [1]◉*, **Jian Chen** [2]◉, **Yixiu Liu** [1], **Jing Lin** [2], **Kun Liu** [1], **Haizhu Chen** [3], **Xuejing Jiang** [1], **Yingjie Zhang** [2], **Xuan Chen** [4], **Binglin Cui** [2], **Shaoping Jiang** [5], **Jianchang Jiang** [6], **Hua Zhang** [7], **Huiyi Hu** [6], **Chendong Li** [1,8], **Wenjuan Li** [9], **E. Li** [3], **Hui Pan** [3]◉*

1 Department of Community Health Sciences, University of Manitoba, Winnipeg, Canada, 2 First Affiliated Hospital of Shantou University Medical College, Shantou, Guangdong, China, 3 Shantou Longhu People's Hospital, Shantou, Guangdong, China, 4 Manitoba Health, Seniors and Active Living, Government of Manitoba, Winnipeg, Canada, 5 School of Mathematics and Statistics, Yunnan Minzu University, Kunming, China, 6 Affiliated Nanhai Hospital of Southern Medical University, Foshan, Guangdong, China, 7 School of Public Health, Southeast University, Nanjing, China, 8 School of Sciences, Nanjing Forest University, Nanjing, China, 9 AstraZeneca China Co. Ltd, Shanghai, China

◉ These authors contributed equally to this work.
* depeng.jiang@umanitoba.ca (DJ); michaelhuipan@163.com (HP)

**Data Availability Statement:** This was an anonymous survey, and confidentiality of data was ensured. Data cannot be shared publicly because

## Abstract

### Background

The COVID-19 pandemic has alarming implications for individual and population level mental health. Although the future of COVID-19 is unknown at present, more countries or regions start to ease restrictions. The findings from this study have provided the empirical evidence of prevalence and patterns of mental disorders in Chinese general population before and after easing most COVID-19 restrictions, and information of the factors associated with these patterns.

### Methods

A cross-sectional population-based online survey was carried out from February to March 2020 in the general population across all provinces in China. The 12-item General Health Questionnaire (GHQ-12) was incorporated in the survey. Latent class analyses were performed to investigate the patterns of mental disorders and multinomial logistic regressions were used to examine how individual and regional risk factors can predict mental disorder patterns.

### Results

Four distinctive patterns of mental health were revealed in the general population. After the ease of most COVID-19 restrictions, the prevalence of high risk of mental disorders decreased from 25.8% to 20.9% and prevalence of being high risk of unhappiness and loss of confidence decreased from 10.1% to 8.1%. However, the prevalence of stressed, social

of the confidential information. The de-identified data are available for researchers who meet the criterial for access to confidential data. Requests for the de-identified data should be sent to Dr. Xianyou Chen, from the Ethical Committee of Shantou Longhu People's Hospital, Shantou, Guangdong, China Email: 546645316@qq.com.

**Funding:** This project was partially funded by the Li Ka Shing Foundation (378073, 2020; HP) and Canadian Institute of Health Research (378073, 2016; DJ). The fund was mostly used for lucky lottery draw and a symbolic thanks of the participation of the survey. The funders had no role in study design, data collection and analysis, decision to publish, or preparation of the manuscript. No author had received a salary from any of these funders.

**Competing interests:** No authors, including Wenjuan Li from Yunque Medical Technology Shanghai Co. Ltd who just moved to a new position at AstraZeneca China Co. Ltd, have competing interests. The funders, Li Ka Shing Foundation and Canadian Institute of Health Research, have not provided any salaries for any author, and have no role in study design, data collection and analysis, decision to publish, or preparation of the manuscript. The commercial affiliation (AstraZeneca China Co. Ltd) does not alter our adherence to PLOS ONE policies on sharing data and materials.

dysfunction and unhappy were consistently high before and after easing restrictions. Several regional factors, such as case mortality rate and healthcare resources, were associated with mental health status. Of note, healthcare workers were less likely to have mental disorders, compared to other professionals and students.

## Conclusions

The dynamic management of mental health and psychosocial well-being is as important as that of physical health both before and after the ease of COVID-19 restrictions. Our findings may help in mental health interventions in other countries and regions while easing COVID-19 restrictions.

## Introduction

Since China reported its first cases to the World Health Organization (WHO) in December 2019, over 140 million COVID-19 cases had been reported worldwide, with more than 3 million deaths by late April 2021 [1]. At least 200 countries have implemented varying degrees of restrictions on population movement to contain the spread of novel coronavirus disease 2019 (COVID-19). While these interventions may be critical in mitigating the spread of the disease during the pandemic crisis, they have generated and intensified stress, as well as negatively affected mental health and well-beings of general population [2, 3]. The symptoms of COVID-19-associated mental health problems include a large range of emotional and behavioral reactions, such as social dysfunction, loss of confidence, depression, anxiety, and insomnia [4–7].

Before the outbreak, the prevalence of mental disorders (excluding dementia) in general population in China was 9.3% (95% CI: 5.4–13.3) in a 12-month study (Huang et al., 2019) [8]. About 16% and 13% of general population had a mood disorder and an anxiety disorder, respectively [9]. The pooled prevalence of insomnia in general population was 15.0% (95% CI: 12.1–18.5) in China according to a meta-analysis in 2017 [10]. Two national surveys [7] conducted in China right before and during the COVID-19 outbreak reported that the pandemic led to a 74% drop in overall emotional wellbeing. Several cross-sectional surveys on the public during the early stages of the outbreak found high levels of mental health problems, and increased symptoms of depression, anxiety, and stress related to COVID-19 [4, 5, 11].

There can be various risk factors associated with mental health status [11, 12]. Sociodemographic factors, such as gender, age, marital status, and Grade 12 graduation, have been reported as essential components [11, 12]. Another important factor is the presence of the peak in the epi curve. After the peak, the daily confirmed cases start to level off or decline, which provides great relief for public health. Several regional factors such as the capability of medical supports and resources and the severity level of the COVID-19 were associated with mental health [13, 14]. The COVID-19 pandemic has resulted in an increase in known risk factors for mental health. Frontline healthcare workers working under extreme conditions and at high risk of getting infected have been experiencing more psychological burdens during this pandemic [15–17]. People with underlying medical conditions are under unprecedented pressure and are experiencing severe psychological distress due to the limited resources for testing and treatment, restrictions/lockdowns, and financial losses [14, 18].

Most previous investigations on COVID-19 related mental health focused on specific subgroups of the population and only a few studies focused on the general populations. A systematic review [19] revealed that studies on general population reported quite different prevalence rates

of psychological distress because of various measurement scales, different reporting patterns, and possibly international/cultural differences. As of February 25, 2020, the State Council of the People's Republic of China declared that the disease transmission had been under control and eased most COVID-19 restrictions in many regions (we consider this as an indicator for easing restrictions. After that, the number of new confirmed COVID-19 cases in mainland China, excluding Hubei province (i.e., the province most severely affected by COVID-19 in China), decreased to under ten for the first time. To our knowledge, no studies have been conducted to examine the differences before and after easing restrictions in the general public's psychological health. The objectives of the study were to provide empirical evidence of the patterns of mental health in general population both before and after easing restrictions and to examine the factors at both individual and regional levels that contributed to and/or mitigated these patterns.

## Materials and methods

### Design and population

A cross-sectional and large-scale survey on physical and mental health conditions of Chinese general population along with their medical care needs and knowledge about the COVID-19 was carried out from February 18 to March 12, 2020. An online questionnaire was circulated via WeChat, a most popular social media platform, to collect information among participants from mainland China, and other regions/countries. This was an anonymous survey and the confidentiality of data was ensured. This study was approved by the Ethical Committee of Shantou Longhu People's Hospital, Shantou, Guangdong, China.

**Procedures.** The link to the questionnaire was posted and re-posted to multiple WeChat groups and WeChat Moments as a snowballing method. The electronic informed consent was obtained prior to starting the questionnaire from each participant, or his/her parent for those who was younger than 18 years. They could choose either to complete the survey or opt out at any time. Each WeChat account owner was limited to submit only one response. The survey data were stored in the server of Wenjuanxing platform and could be accessed only by the authorized researchers from the involved organizations/institutions.

**Exclusion criteria.** Exclusion criteria were set as follows: a) being aged < 12 years; b) being aged < 20 years and being married, divorced or widowed; c) being aged < 20 years and having a degree of Master or PhD; or d) questionnaires completed in $\leq$ 50 seconds. Exclusion criteria b) and c) were set because these participants failed to pass the internal consistent checks. Those participants, who were younger than 12 years or completed questionnaires in $\leq$ 50 seconds, were also excluded for the data quality control reasons.

**Measurements.** The 12-item General Health Questionnaire (GHQ-12) was incorporated in the survey to evaluate the participants' mental well-being. The General Health Questionnaire (GHQ) has been extensively used as a psychiatric disorder screening tool. The GHQ-12 is the shortest questionnaire amongst the GHQ series yet that offers comparable screening accuracy [20, 21]. The reliability and validity of GHQ-12 have been examined in many countries, including China, and reported appropriate to use [22, 23]. Determining the cut-off points for the GHQ-12 scores is challenging and varies according to regions, populations, and the time of a study [24, 25]. Instead of using the traditional scoring method of GHQ-12, we used the Latent Class Analysis (LCA) [26] to investigate the patterns and prevalence of mental disorders in general population during the COVID-19 pandemic.

### Statistical analysis

The three stages of data analysis can be described as follows. In the first stage, LCA was used to investigate the patterns of mental disorders for the participants before easing restrictions and

after easing restrictions separately. LCA was also used to classify them into distinct classes. Individuals classified into the same class are similar to each other and different from those in other classes. In the second stage, the prevalence of each pattern of mental disorders for each sociodemographic and disease group was estimated by using multinomial logistic regressions. In the final stage, the multivariable multinomial logistic regressions were conducted to examine how individual and regional factors predicted mental disorder patterns. All multinomial logistic regressions were conducted on cohorts before and after easing restrictions separately.

## Results

Out of 430,152 visits to the questionnaire from early February to mid-March, 2020, 108,218 individuals completed it (response rate: 25.16%). Final samples available for analyses included 46,508 participants before easing restrictions and 58,740 participants after easing restrictions. Of the total sample, 57,262 (54.4%) were male, and the mean (SD) age was 30.0 (9.8) years with a range of 12 to 100 years. 64,030 participants (60.8%) had a college degree or higher, and 57,999 (53.2%) were married. 16,049 (15.5%) participants were healthcare workers including doctors, nursing professionals, midwifery professionals, dentists and pharmacists, and 19,738 (18.7%) were unemployed. 819 (0.8%) participants had at least one respiratory disease including pneumonia, asthma, and COPD, and 3,206 (3.1%) had one or more non-respiratory diseases.

LCA was conducted for the two cohorts (before and after easing restrictions) separately. As in most LCAs, the bimodal GHQ score method (item as 0-1-1-1) was used for each item in this study, which served as indicators for LCA. We successively tested several models in an iterative fashion to determine the model with the optimal number of classes. A four-class solution was chosen for both cohorts based on Bayesian Information Criterion (BIC), Lo Mendell Rubin likelihood ratio test, and entropy value [27].

The four-class solutions also exhibited good clinical interpretability. As presented in Fig 1, the symptom endorsement profiles of participants were highly comparable across the four classes and the risk profiles were quite similar before and after easing restrictions. Participants in the high-risk group (Class One) displayed high probabilities of all 12 mental disorder indicators (0.80–0.95). This high-risk class was estimated to account for 25.8% of participants before easing restrictions and 20.9% of participants after easing restrictions. Class Two, representing 28.4% of the sample before easing restrictions and 32.8% of the sample after easing restrictions, was identified as "stressed, social dysfunction, and unhappy" class. Participants in Class two had higher probabilities of being unhappy, unable to concentrate and under strain (range of probabilities: [0.43–0.63] and elevated risk on other indicators [0.21–0.38]). Class Three, representing 10.1% of the sample before easing restrictions and 8.1% of the sample after easing restrictions, was identified as "unhappy and loss of confidence" class. Participants in Class Three had higher probabilities of being unhappy and loss of confidence (range of probabilities: 0.68–0.91; and intermediate probabilities of other indicators: 0.11–0.51). The largest class (35.7% of the sample before and 38.2% of the sample after easing restrictions), so-called low-risk class, was comprised of participants who had low or zero probabilities for all mental disorder indicators.

Participants were assigned to a latent class based on their highest estimated posterior class probability. A series of univariate multinomial logistic regression were conducted separately for two cohorts to examine the prevalence of the four risk classes for each sociodemographic and/or disease group. Before easing restrictions, as displayed in Table 1, males had a higher chance for being in "High Risk" (Class One), whereas females had a higher likelihood for belonging to "Stressed, Social Dysfunction and Unhappy" (Class Two). However, after easing

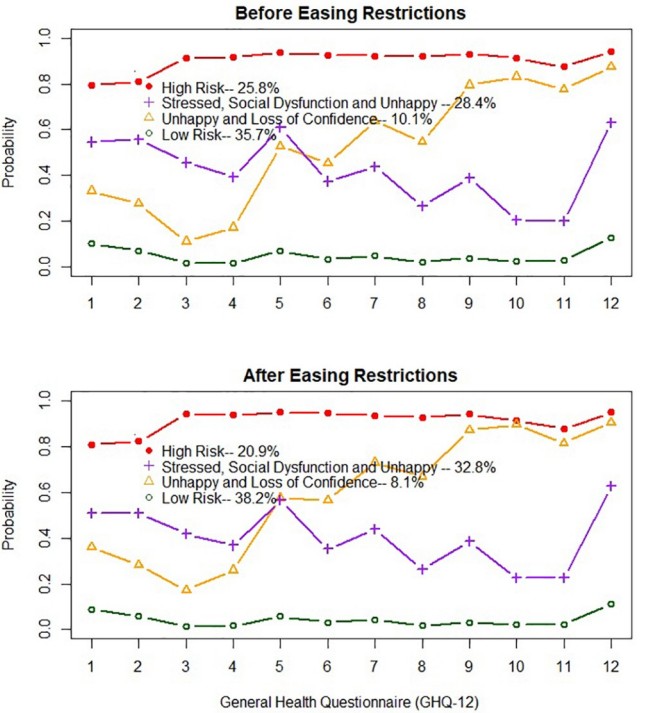

**Fig 1. Profile of mental health problems.** 1. Unable to concentrate; 2. Loss of sleep; 3. Play much less useful part; 4. Unable to make decisions; 5. Under strain; 6. Couldn't overcome difficulties; 7. Unable to enjoy 8. Unable to face up problems; 9. Unhappy and depressed; 10. Lose confidence; 11. Worthless person; 12. Feel unhappy.

**Table 1. Prevalence of mental health problems by characteristic—Before easing COVID-19 restrictions (N = 46508).**

| Characteristic | Groups | N | High Risk | Stressed, Social Dysfunction and Unhappy | Unhappy and Loss of Confidence |
|---|---|---|---|---|---|
| | | | % (95% CI) | % (95% CI) | %(95% CI) |
| Gender | Female | 19188 | 24.7(24.1–25.3) | 31.4(30.7–32.1) | 9.8(9.4–10.2) |
| | Male | 27320 | 26.5(26.0–27.1) | 26.2(25.7–26.7) | 10.4(10.0–10.7) |
| Marriage status | Married | 23043 | 23.5(23.0–24.1) | 30.2(29.6–30.8) | 8.8(6.9–10.7) |
| | Divorced/Widowed | 839 | 51.6(48.3–55.0) | 21.5(28.7–34.3) | 8.8(8.4–9.1) |
| | Single or never married | 22626 | 27.1(26.6–27.7) | 26.8(26.2–27.3) | 11.6(11.1–12.0) |
| Education | High school or below | 16862 | 27.5(26.9–28.2) | 28.3(37.6–29.0) | 10.4(9.9–10.8) |
| | University/college or above | 29646 | 24.8(24.3–25.3) | 28.4(27.9–28.9) | 10.0(9.7–10.3) |
| Occupation | HWs[a] | 7792 | 21.9(21.0–22.9) | 26.1(25.1–27.1) | 10.2(9.5–10.9) |
| | Non-HWs | 30916 | 27.3(26.8–27.8) | 27.8(27.3–28.3) | 9.8(9.5–10.2) |
| | Unemployed | 7800 | 23.8(22.8–24.7) | 32.7(31.7–33.7) | 11.3(10.6–12.0) |
| Disease status | No | 44697 | 24.7(24.3–25.1) | 28.6(28.2–29.0) | 10.1(9.8–10.4) |
| | Respiratory[b] | 101 | 54.5(44.7–64.2) | 22.8(14.6–31.0) | 8.9(3.4–14.5) |
| | Non-respiratory[c] | 1319 | 49.8(47.1–52.5) | 24.8(22.5–27.1) | 11.1(9.4–12.8) |
| | Both | 391 | 65.7(61.0–70.4) | 16.4(12.7–20.0) | 7.7(5.0–10.3) |
| Perceived needs met | Yes | 44459 | 24.2(23.8–24.6) | 28.7(28.2–29.1) | 10.1(9.8–10.3) |
| | No | 2049 | 60.3(58.2–62.4) | 21.8(20.0–23.6) | 11.7(10.3–13.1) |

Note:

[a] Healthcare workers including doctors, nursing professionals, midwifery professionals, dentists and pharmacists;

[b] Respiratory diseases including pneumonia, asthma, and COPD;

[c] Non-respiratory disease including hypertension, diabetes, heart disease, stroke, hepatitis, cancer, or esophagitis, gastritis, or duodenitis or other.

**Table 2. Prevalence of mental health problems by characteristic—After easing COVID-19 restrictions (N = 58740).**

| Characteristic | Groups | N | High Risk | Stressed, Social Dysfunction and Unhappy | Unhappy and Loss of Confidence |
|---|---|---|---|---|---|
| | | | % (95% CI) | % (95% CI) | % (95% CI) |
| Gender | Female | 28798 | 21.4(20.9–21.8) | 34.0(33.4–34.5) | 8.0(7.7–8.3) |
| | Male | 29942 | 20.6(20.1–21.0) | 31.7(31.2–32.2) | 8.2(7.9–8.5) |
| Marriage status | Married | 34956 | 20.0(19.6–20.5) | 33.3(32.8–33.8) | 7.2(6.9–7.5) |
| | Divorced/Widowed | 1266 | 33.3(30.7–35.9) | 32.7(30.1–35.3) | 7.3(5.9–8.8) |
| | Single or never married | 22518 | 21.7(21.2–22.3) | 32.0(31.4–32.6) | 9.6(9.2–9.9) |
| Education | High school or below | 24356 | 22.4(21.9–22.9) | 32.8(32.2–33.4) | 8.0(7.7–8.4) |
| | University/college or above | 34384 | 19.9(19.5–20.4) | 32.8(32.3–33.3) | 8.1(7.9–8.4) |
| Occupation | HWs[a] | 8257 | 15.3(14.5–16.1) | 27.8(26.8–28.8) | 8.0(7.4–8.6) |
| | Non-HWs | 38545 | 22.2(21.8–22.7) | 32.9(32.4–33.3) | 8.0(7.7–8.2) |
| | Unemployed | 11938 | 20.8(20.1–21.5) | 36.1(35.2–36.9) | 8.6(8.1–9.1) |
| Disease status | No | 57163 | 20.4(20.1–20.8) | 32.8(32.4–33.2) | 8.0(7.8–8.3) |
| | Respiratory[b] | 81 | 32.1(21.9–42.3) | 34.5(24.2–44.9) | 12.4(5.2–19.5) |
| | Non-respiratory[c] | 1250 | 39.0(36.3–41.7) | 33.2(30.6–35.8) | 10.5(8.8–12.2) |
| | Both | 246 | 50.0(43.7–56.2) | 27.7(22.1–33.2) | 8.1(4.7–11.6) |
| Perceived needs met | Yes | 56750 | 20.1(19.8–20.5) | 32.7(32.3–33.1) | 8.0(7.7–8.2) |
| | No | 1990 | 44.6(42.4–46.8) | 35.8(33.7–37.9) | 12.1(10.6–13.5) |

Note:

[a] Healthcare workers including doctors, nursing professionals, midwifery professionals, dentists and pharmacists;

[b] Respiratory diseases including pneumonia, asthma, and COPD;

[c] Non-respiratory disease including hypertension, diabetes, heart disease, stroke, hepatitis, cancer, or esophagitis, gastritis, or duodenitis or other.

restrictions as shown in Table 2, females had a greater chance of being in these two disorder classes. The prevalence of "High Risk" (Class One) was much higher in the divorced or widowed than others, the prevalence rate of "Stressed, Social Dysfunction and Unhappy" (Class Two) was higher for the married, while the single or unmarried had higher probability being "Unhappy and Loss of Confidence" (Class Three). The prevalence of "High Risk" (Class One) was higher among those with high school or lower education levels than those with college or above education levels. The prevalence of Classes One and Two was lower among healthcare workers than people in the other occupations. Participants with respiratory diseases and/or other chronic diseases had a greater chance of being in those three disorder classes (i.e., Classes One, Two and Three) than healthy participants. These patterns were consistent before and after easing restrictions.

In the final analysis, the multivariable multinomial logistic regression was conducted separately for two cohorts to examine how individual and regional factors could predict mental disorder patterns. The parameter estimates and adjusted odds ratios for each mental disorder class by each predictor are shows in Tables 3 and 4, respectively. Females had a higher likelihood of belonging to any of the mental disorder risk classes. In particular, females had increased odds of belonging to Class Two ("Stressed, Social Dysfunction and Unhappy"). In addition, the following participants also had increased likelihood of being in any of the mental disorder risk classes (Classes One, Two and Three): seniors, teenagers, single/unmarried or divorced/widowed participants, the participants with diseases or perceived unmet medical care needs, and those from provinces of higher case fatality rates or lower temperature. Low education increased the likelihood of association in the "High Risk" class. In comparison with other professionals or students, the healthcare workers were less likely to belong to any of the

**Table 3. Parameter estimates (standard errors) from fitted multinomial logistic regression predicting mental health profiles.**

| | Before Easing COVID-19 Restrictions | | | After Easing COVID-19 Restrictions | | |
|---|---|---|---|---|---|---|
| | High Risk | Stressed, Social Dysfunction and Unhappy | Unhappy and Loss of Confidence | High Risk | Stressed, Social Dysfunction and Unhappy | Unhappy and Loss of Confidence |
| Intercept | 2.61 (0.15)*** | 0.80(0.15)*** | 0.19(0.19) | 1.74 (0.14)*** | 0.66(0.13)*** | -0.05(0.17) |
| Demographics | | | | | | |
| Male | -0.08 (0.03)** | -0.24(0.02)*** | -0.09(0.03)** | -0.15 (0.02)** | -0.15(0.02)*** | -0.12(0.03)** |
| Age | -0.12 (0.02)*** | 0.08(0.02)*** | -0.04(0.03) | -0.10 (0.02)*** | 0.01(0.02) | -0.04(0.03) |
| Age*Age | 0.06 (0.01)*** | -0.02(0.01)+ | 0.02(0.01)* | 0.04 (0.01)*** | 0.01(0.01)+ | 0.02(0.02) |
| Marriage Status(ref = 'Married') | | | | | | |
| Divorced/Widowed | 1.21 (0.10)*** | 0.35(0.11)** | 0.64(0.15)*** | 0.72 (0.08)*** | 0.32(0.08)*** | 0.35(0.12)** |
| Single or never married | 0.12 (0.04)** | 0.05(0.03) | 0.27(0.05)*** | 0.05 (0.03) | 0.02(0.03) | 0.26(0.05)*** |
| High school or below | 0.07 (0.03)** | 0.03(0.03) | 0.04(0.04) | 0.10 (0.02)*** | 0.00(0.02) | 0.01(0.03) |
| Occupation (ref = 'Non HWs') | | | | | | |
| HWs | -0.44 (0.03)*** | -0.25(0.03)*** | -0.17(0.05)*** | -0.70 (0.04)*** | -0.46(0.03)*** | -0.32(0.05)*** |
| Unemployed | -0.13 (0.04)*** | 0.17(0.03)*** | 0.04(0.05) | -0.10 (0.03)*** | 0.09(0.03)*** | 0.02(0.04) |
| Days since Peak | -0.008 (0.005)+ | -0.015(0.004)** | -0.005(0.006) | 0.013 (0.004)*** | -0.014(0.003)*** | 0.009(0.005)+ |
| Physical Conditions (ref = 'No Disease') | | | | | | |
| Respiratory only | 1.53 (0.31)*** | 0.76(0.34)* | 0.79(0.43)+ | 0.74(0.32)* | 0.51(0.31)+ | 0.83(0.40)* |
| Non-Respiratory only | 1.50 (0.09)*** | 0.78(0.09)*** | 1.02(0.11)*** | 1.35 (0.08)*** | 0.77(0.09)*** | 1.03(0.11)*** |
| Both | 2.12 (0.17)*** | 0.82(0.20)*** | 1.10(0.24)*** | 1.82 (0.20)*** | 0.91(0.21)*** | 1.10(0.28)*** |
| Perceived needs met | -2.58 (0.09)*** | -1.49(0.10)*** | -1.90(0.11)*** | -2.38 (0.09)*** | -1.72(0.09)*** | -2.03(0.11)*** |
| Case fatality rate | -0.02(0.02) | 0.06(0.02)** | 0.04(0.03) | 0.07(0.02)** | 0.07(0.02)*** | 0.05(0.03)+ |
| Number of beds | -0.10 (0.02)*** | 0.10(0.02)*** | 0.05(0.03)+ | -0.01(0.02) | 0.18(0.02)*** | 0.07(0.03)* |
| Lowest temperature | 0.00(0.002) | 0.02(0.002)*** | 0.03(0.003)*** | 0.008 (0.002)*** | 0.03(0.002)*** | 0.03(0.003)*** |

Note:

+ $p < .10$,

* $p < .05$,

** $p < .01$,

*** $p < .001$.

Reference category is 'Low Risk'.

**Table 4. Odds ratios (95% CI) from fitted multinomial logistic regression predicting mental health profiles.**

| | Before | | | After | | |
|---|---|---|---|---|---|---|
| | High Risk | Stressed, Social Dysfunction and Unhappy | Unhappy and Loss of Confidence | High Risk | Stressed, Social Dysfunction and Unhappy | Unhappy and Loss of Confidence |
| Demographics | | | | | | |
| Male | 0.92(0.88–0.97) | 0.79(0.75–0.83) | 0.91(0.85–0.98) | 0.86(0.82–0.90) | 0.85(0.82–0.89) | 0.89(0.83–0.95) |
| Marriage Status(ref = 'Married') | | | | | | |
| Divorced/Widowed | 3.37(2.77–4.10) | 1.42(1.14–1.77) | 1.89(1.42–2.51) | 2.06(1.77–2.40) | 1.38(1.19–1.60) | 1.43(1.13–1.81) |
| Single or never married | 1.13(1.05–1.21) | 1.05(0.98–1.12) | 1.31(1.20–1.44) | 1.05(0.99–1.12) | 1.02(0.97–1.08) | 1.29(1.18–1.41) |
| High school or below | 1.07(1.02–1.13) | 1.03(0.98–1.09) | 1.04(0.97–1.12) | 1.11(1.05–1.06) | 1.00(0.96–1.04) | 1.01(0.94–1.08) |
| Occupation (ref = 'Non HWs') | | | | | | |
| HWs | 0.64(0.60–0.69) | 0.78(0.73–0.83) | 0.84(0.77–0.92) | 0.50(0.46–0.53) | 0.63(0.60–0.67) | 0.73(0.67–0.80) |
| Unemployed | 0.88(0.82–0.95) | 1.18(1.11–1.26) | 1.05(0.96–1.15) | 0.90(0.85–0.96) | 1.09(1.04–1.15) | 1.02(0.94–1.11) |
| Days since Peak | 0.99(0.98–1.00) | 0.99(0.98–0.99) | 1.00(0.98–1.01) | 1.01(1.01–1.02) | 0.99(0.98–0.99) | 1.01(0.99–1.02) |
| Physical Conditions (ref = 'No Disease') | | | | | | |
| Respiratory only | 4.61(2.52–8.46) | 2.14(1.10–4.17) | 2.20(0.95–5.10) | 2.11(1.12–3.97) | 1.67(0.91–3.09) | 2.30(1.04–5.08) |
| Non-Respiratory only | 4.47(3.77–5.28) | 2.17(1.81–2.61) | 2.78(2.23–3.46) | 3.87(3.27–4.56) | 2.17(1.83–2.56) | 2.80(2.24–3.49) |
| Both | 8.34(5.93–11.7) | 2.27(1.53–3.38) | 3.02(1.87–4.86) | 6.15(4.18–9.07) | 2.49(1.65–3.76) | 2.99(1.72–5.21) |
| Perceived needs met | 0.08(0.06–0.09) | 0.22(0.18–0.27) | 0.15(0.12–0.19) | 0.09(0.08–0.11) | 0.18(0.15–0.22) | 0.13(0.11–0.16) |
| Case fatality rate | 0.98(0.94–1.03) | 1.06(1.01–1.10) | 1.04(0.98–1.10) | 1.07(1.03–1.11) | 1.08(1.04–1.12) | 1.05(1.00–1.12) |
| Number of beds | 0.91(0.87–0.95) | 1.10(1.06–1.15) | 1.05(0.99–1.10) | 0.99(0.95–1.03) | 1.20(1.16–1.24) | 1.07(1.02–1.13) |
| Lowest temperature | 1.00(1.00–1.01) | 1.03(1.02–1.03) | 1.03(1.02–1.03) | 1.01(1.00–1.01) | 1.03(1.025–1.032) | 1.03(1.02–1.04) |

Note: Reference category is 'Low Risk'.

mental disorder risk classes, and unemployed participants were more likely to belong to the "Stressed, Social Dysfunction and Unhappy" class (Class Two), but less likely to belong to the "High Risk" class (Class One). The regions with a higher number of hospital beds had a lower prevalence rate of "High Risk", whereas participants from provinces with a higher number of hospital beds were more likely to belong to Classes Two and Three. The likelihood of being "Stressed, Social Dysfunction and Unhappy" decreased day by day after the peak of cases. The above associations between these risk factors and mental disorders are quite similar before and after easing restrictions.

## Discussion

The current study has examined the patterns of mental health disorders and associated factors among the general population in China during the COVID-19 pandemic. Based on a survey over 100,000 participants across all provinces/regions in China, results from latent class analysis revealed that more than one-fifth of the general population were at high risk of mental disorders with symptoms as being stressed, being unhappy, loss of confidence, and social dysfunction. Almost one third of the participants were at moderate risk of being unhappy, being stressed, and social dysfunction; one tenth was at moderate to high risk of unhappiness and loss of confidence. The prevalence of mental health symptoms differed significantly by stages of outbreak. After the ease of most COVID-19 restrictions, the prevalence of high risk of

mental disorders decreased from 25.8% to 20.9% and the prevalence of being high risk of unhappiness and loss of confidence decreased from 10.1% to 8.1%. However, the prevalence of stressed, social dysfunction and unhappy are consistently high before and after easing restrictions.

The highlight of this study was to explore factors that contributed to, or mitigated these mental problems, as well as to identify the key populations that should be set as a priority for psychological interventions. Our results indicate that the participants living with one or more chronic diseases were three or four times more likely at risk of mental health disorders. Participants with multiple chronic conditions, especially co-occurring respiratory disease(s), were seven or eight times more likely at risk of mental health problems. Therefore, considerations should be made for people with pre-existing chronic diseases whose care might be disrupted during the COVID-19 pandemic. Steps should be taken to ensure that these people have access to medications without interruptions during the pandemic.

We have also found that the divorced, widowed, or single participants tended to have a higher level of mental health problems. Divorced or widowed participants were two or three times more likely to have psychological disorders and those single or unmarried ones had elevated odds of being unhappy or loss of confidence. Having less communications or supports could be a reason behind this. Provision of more and better mental/social supports may promote mental health of the vulnerable during the pandemic while keeping physical distance.

In other studies [20, 28], healthcare workers have reported negative consequences as a result of stress exposure and fear of getting infected or infecting their families and friends. However, our healthcare workers were less likely to be at high risk of mental disorders, compared to other professionals and students. This might be due to their better knowledge of the disease, protective measures, and professional trainings. The survey was conducted in late February when healthcare workers had been provided with personal protective equipment and psychosocial supports. Another possible reason could be that those frontline healthcare workers were too busy to respond to our survey.

Our results have shown that teenagers or young adults and seniors were vulnerable to mental health/emotional problems. The teenagers or young adults might be at particular risk during the pandemic as quarantined children were more likely to develop acute stress disorder, adjusted disorders, and grief [29]. Elderly people were as well at high risk of having severe COVID-19 illness and mental-health-related consequences because they might already have some cognitive decline [30]. Special considerations should be made to ensure that local community health services, such as schools, community centers for the youths and seniors, should be continued to carry out regular services during the pandemic.

The strengths of this study include its huge sample size, extensive geographic coverage across China, the special study period and the use of advanced statistical techniques. The survey covered both period before and after the ease of most COVID-19 restrictions. We also adjusted for individual and regional factors as well as the stage of pandemic.

Several limitations of this study are worth noting. This survey was based on a convenience sampling methods and the sample might not be representative for certain groups such as frontline healthcare workers and non-internet users. Those with serious mental disorders may be less likely to participate in the survey, and those in particular regions may be more or less likely to participate. Future studies should recruit a representative probability sample or use other social media such as Weibo in order to draw more reliable conclusions. Current study with a cross-sectional design could not evaluate long-term consequences of COVID-19 on mental health. The sample sets used in the two stages before and after the easing of restrictions were quite different in demographics (see S1 Table). Therefore, association between the ease of restrictions and mental health patterns cannot necessarily be considered causal relationships.

The survey was fielded in February and March 2020 when the situation of the pandemic was dramatically different from the other periods of the year and early 2021. Thus the prevalence and patterns of mental disorder might not apply to other pandemic periods. The longitudinal studies with follow-up assessments at different periods of pandemic are needed to determine the transition of mental health patterns and the long-term mental health outcomes.

In summary, COVID-19 is both magnifying and contributing to the patterns of mental health disorders in general population. As more countries start to ease some COVID-19 restrictions, it is essential to identify the patterns of mental disorders among different populations and different stages. Some groups (e.g., with pre-existing chronic diseases including mental health problems) are at greater risk of developing more severe difficulties. The capacity of seeing a psychiatrist/psychologist/social worker will be critical for them. Understanding and addressing mental health and psychosocial concerns will be one of the key steps to break down disease transmission, to prevent long-term repercussions on the population's wellbeing, and to improve their ability to cope with adversity and stress. Mental health interventions should be carried out within general health services (including primary health care). Communities and organizations could consider training nontraditional groups to provide psychological first aids, and mental health clinicians should work with these groups to develop standardized, evidence-informed resources. Governments should strengthen legislations to improve workplace mental health and provide incentives to employers for implementing robust mental health strategies. Although the future of COVID-19 is unknown at present, the dynamic management of mental health and psychosocial well-being is as important as that of physical health both before and after the ease of restrictions.

## Supporting information

**S1 Fig. Number of valid sample size by collection date.**
(TIF)

**S1 Table. Sample characteristics of survey participants before and after easing restrictions.**
(DOCX)

## Author Contributions

**Conceptualization:** Depeng Jiang, Jian Chen, Jing Lin, Haizhu Chen, Yingjie Zhang, Xuan Chen, Binglin Cui, Jianchang Jiang, Hua Zhang, Huiyi Hu, Chendong Li, Wenjuan Li, E. Li, Hui Pan.

**Data curation:** Hui Pan.

**Formal analysis:** Depeng Jiang, Yixiu Liu, Kun Liu, Shaoping Jiang.

**Funding acquisition:** Depeng Jiang, Hui Pan.

**Investigation:** Depeng Jiang, Hui Pan.

**Methodology:** Depeng Jiang, Yixiu Liu, Kun Liu, Xuejing Jiang, Xuan Chen, Chendong Li, Hui Pan.

**Project administration:** Depeng Jiang, Hui Pan.

**Resources:** Depeng Jiang.

**Software:** Depeng Jiang.

**Supervision:** Depeng Jiang, Jian Chen, Jing Lin, Hui Pan.

**Validation:** Depeng Jiang.

**Visualization:** Depeng Jiang, Kun Liu, Shaoping Jiang, Hua Zhang.

**Writing – original draft:** Depeng Jiang, Yixiu Liu, Kun Liu.

**Writing – review & editing:** Depeng Jiang, Jian Chen, Jing Lin, Kun Liu, Haizhu Chen, Xuejing Jiang, Yingjie Zhang, Xuan Chen, Binglin Cui, Shaoping Jiang, Jianchang Jiang, Hua Zhang, Huiyi Hu, Chendong Li, Wenjuan Li, E. Li, Hui Pan.

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
