## [Decision Letter · Decision Letter 0]

9 Jun 2021

PONE-D-21-14417

Patterns of Mental Health Problems Before and After Easing COVID-19 Restrictions: Evidence from a 105248-subject Survey in General Population in China

PLOS ONE

Dear Dr. Jiang,

Thank you for submitting your manuscript to PLOS ONE. After careful consideration, we feel that it has merit but does not fully meet PLOS ONE’s publication criteria as it currently stands. Therefore, we invite you to submit a revised version of the manuscript that addresses the points raised during the review process.

We look forward to receiving your revised manuscript.

Kind regards,

Wen-Jun Tu

Academic Editor

PLOS ONE

Journal Requirements:

"No authors have competing interests."

We note that one or more of the authors are employed by a commercial company: Yunque Medical Technology Shanghai Co. Ltd.

4.1. Please provide an amended Funding Statement declaring this commercial affiliation, as well as a statement regarding the Role of Funders in your study. If the funding organization did not play a role in the study design, data collection and analysis, decision to publish, or preparation of the manuscript and only provided financial support in the form of authors' salaries and/or research materials, please review your statements relating to the author contributions, and ensure you have specifically and accurately indicated the role(s) that these authors had in your study. You can update author roles in the Author Contributions section of the online submission form.

4.2. Please also provide an updated Competing Interests Statement declaring this commercial affiliation along with any other relevant declarations relating to employment, consultancy, patents, products in development, or marketed products, etc.  

Reviewers' comments:

Reviewer's Responses to Questions

**Comments to the Author**

1. Is the manuscript technically sound, and do the data support the conclusions?

Reviewer #1: Yes

Reviewer #2: Yes

2. Has the statistical analysis been performed appropriately and rigorously? 

Reviewer #1: No

Reviewer #2: Yes

3. Have the authors made all data underlying the findings in their manuscript fully available?

Reviewer #1: Yes

Reviewer #2: Yes

4. Is the manuscript presented in an intelligible fashion and written in standard English?

Reviewer #1: Yes

Reviewer #2: Yes

5. Review Comments to the Author

Reviewer #1: -A single Wenjuanxing platform was used for data collection. A single platform may have a biased sample due to user groups. Therefore, the rationality of using a single Wenjuanxing platform and the potential impact on the sample should be explained.

-The time of data collection needs to be given more precisely. Due to the long collection time, a trend graph of data collection can be drawn.

-Different sample sets were used in the two stages before and after. It may be due to changes in platform distribution methods and other reasons that the sample structure before and after is quite different (such as age, gender, region, etc.), resulting in a lack of comparability in the analysis. Therefore, it is recommended to explain the risk.

-Table 3 does not give regression statistical parameters, such as R2, F value, etc.

Reviewer #2: Review of Manuscript ID: PONE-D-21-14417

Title of manuscript: Patterns of Mental Health Problems Before and After Easing COVID-19 Restrictions: Evidence from a 105248-subject Survey in General Population in China

The overall purposes of this study are to provide empirical evidence of the patterns of mental health in large general population in China both before and after easing restrictions and to examine the factors at both individual and regional levels that contributed to and/or mitigated these patterns. Understanding these patterns is very important for policy makers and health care workers as well as the scholarly community; thus this paper can be very impactful. In addition, the size of the survey data is unusual and appealing in light of the current literature on this topic, which typically relies on the limited number of respondents for empirical analysis. Overall, the paper had a clear message and was very well-written. I think this paper can make a significant contribution to the literature in mental health among general population during pandemic.

Below I have some minor comments that could improve the manuscript.

Abstract:

• Suggest to add brief information about the mental health issues has become a major health concerns during the COVID-19 pandemic.

• Suggestion to change the last sentence in background: The findings from this research study have provided the empirical evidence …………………, and information of the factors associated with these patterns.

Introduction: It is well organized and easy to follow. The authors have provided sufficient up-to-date citations from other studies which support the readers to understand the current research gaps, objectives and the findings from this paper.

• Page 11 ( page #22 in paper), Line 40 : Suggest to change “To our knowledge, no studies have been conducted to examine the …”

• Page 11 (page#22 in the paper), Line 41-43: Suggest to delete commas and change “ The objectives of the study were to provide empirical evidence of the patterns of mental health in general population both before and after easing restrictions and to examine the factors at both individual and regional levels that contributed to and/or mitigated these mental health patterns.

Methods:

• Page 12 ( page #23 in paper), Line 58: Exclusion criteria – the authors may include the reasons why these groups of population (b. being aged < 20 years and being married, divorced or widowed; c. being aged < 20 years and having a degree of Master or PhD) were excluded from this current study.

• Page 13 (page #24 in paper), Line 75: suggest to change “LCA was also used to classify individuals into distinct classes.”

Results:

• Page 14 (page # 25 in paper), Line 116: Suggest to specify which classes. E.g., However, after easing restrictions as shown in Table 2, females had a greater chance of being in both “High Risk (Class one) and Stressed, Social Dysfunction and Unhappy (Class Two)” .

• Page 15: (page #26 in paper), Line 132: Suggest to provide additional information on the age range/group for seniors and teenagers and include them in the methods section.

• Page 17: (page# 28 in paper), Line 178: Suggest to change the word “children” to “teenagers or young adults” because people who are younger than 12 years old were excluded from the study; Several terms were used to represent a “young adults” group, such as children, youths, teenagers, as well as terms for “seniors”, such as elderly people – suggest to keep them consistent throughout this paper.

Discussion:

• Further to the limitations of this current study, it would be helpful to include the future research studies and/or lines of work that should be considered.

Overall: This paper is clear, well founded that provides the essential information on the mental health patterns and associated factors in a large general population both before and after easing COVID-19 restrictions. Given the potential value of the current study, I strongly recommend that the authors will take the above suggestions into consideration and revise the manuscript, in order to improve this interesting research work.

6. PLOS authors have the option to publish the peer review history of their article (what does this mean?). If published, this will include your full peer review and any attached files.

Reviewer #1: No

Reviewer #2: No

---

## [Author Response · Author response to Decision Letter 0]

23 Jun 2021

Reviewer 1:

Issue 1: A single Wenjuanxing platform was used for data collection. A single platform may have a biased sample due to user groups. Therefore, the rationality of using a single Wenjuanxing platform and the potential impact on the sample should be explained.

Response: Thanks for raising this concern. We have acknowledged the limitation of this convenience sampling methods in the Discussion section (Line 262-275, Page 24-25).

Wenjuanxing (Wenjuanxing TechCo. Ltd., Changsha, China) is an authoritative and widely-used online survey platform that was used to develop the questionnaire and manage data in this study [1]–[7]. Supporting by its technology, the survey hosted by Wenjuanxing can be sent out directly through WeChat wallet [3]. Upon completion, the participant could join a lottery (200 prizes of 50-200 RMB (7-28 USD)) and a symbolic ‘CNY lucky money’ of 1 RMB for 80% of the participants that can be transferred to participants’ WeChat wallet. WeChat is one of the most popular social media platforms with above 1.26 billion users out of the 1.44 billion population in China, which shows its important role and potential in facilitating online surveys. The feasibility of this online survey data collection approach using Wenjuanxing and WeChat platforms was elaborated in an article promoting worldwide researchers to conduct online surveys in China [3]. 

Adopting an internet-based sampling can introduce bias through restricting the respondents to internet-users, their relative and friends [3], [7]. As a consequence, the sample tends to be younger and technologically savvy, in turn, limit its representativeness [3]. One study in Swedish found that the sample recruited through internet survey over-represent the population aged between 25 and 65 and underrepresent younger and older populations [8], [9]. Unfortunately, the representativeness of the sampling through WeChat has not been investigated [3]. Therefore, caution requires when disseminate the results from this study. In the future studies, other social media such as Weibo and QQ can be included when recruiting participants to increase the representativeness of the sample through multiple platforms [10]. We also have added that future directions in the discussion session “Future studies should recruit a representative probability sample or use other social media such as Weibo in order to draw more reliable conclusions” (Line 265-267, Page 24-25).

Reference

[1] T. Zhou, T. V. T. Nguyen, J. Zhong, and J. Liu, “A COVID-19 descriptive study of life after lockdown in Wuhan, China,” R. Soc. Open Sci., vol. 7, no. 9, 2020.

[2] K. Huang et al., “Attitudes of Chinese health sciences postgraduate students’ to the use of information and communication technology in global health research,” BMC Med. Educ., vol. 19, no. 1, pp. 1–10, 2019.

[3] B. Mei and G. T. L. Brown, “Conducting Online Surveys in China,” Soc. Sci. Comput. Rev., vol. 36, no. 6, pp. 721–734, 2018.

[4] H. Liqian, “Study on the Perceived Popularity of Tik Tok,” Bangkok University, 2018.

[5] Y. Wang, F. Guo, J. Wei, Y. Zhang, Z. Liu, and Y. Huang, “Knowledge, attitudes and practices in relation to antimicrobial resistance amongst Chinese public health undergraduates,” J. Glob. Antimicrob. Resist., vol. 23, pp. 9–15, 2020.

[6] J. Gao et al., “Mental health problems and social media exposure during COVID-19 outbreak,” PLoS One, vol. 15, no. 4, pp. 1–10, 2020.

[7] H. Luo, Y. Lie, and F. W. Prinzen, “Surveillance of COVID-19 in the general population using an online questionnaire: Report from 18,161 respondents in China,” JMIR Public Heal. Surveill., vol. 6, no. 2, pp. 1–14, 2020.

[8] M. W. Ross, S. A. Månsson, K. Daneback, A. Cooper, and R. Tikkanen, “Biases in internet sexual health samples: Comparison of an internet sexuality survey and a national sexual health survey in Sweden,” Soc. Sci. Med., vol. 61, no. 1, pp. 245–252, 2005.

[9] R. L. Marquet et al., “Internet-based monitoring of influenza-like illness (ILI) in the general population of the Netherlands during the 2003-2004 influenza season,” BMC Public Health, vol. 6, no. Ili, pp. 1–8, 2006.

[10] V. D. De Rada, L. V. C. Ariño, and M. G. Blasco, “The use of online social networks as a promotional tool for self-administered internet surveys,” Rev. Española Sociol., vol. 25, no. 2, pp. 189–203, 2016.

Issue 2: The time of data collection needs to be given more precisely. Due to the long collection time, a trend graph of data collection can be drawn. 

Response: Thanks for the suggestion. We have given the data collection start and end dates in the Design section (Line 94, Page 3). For your information, the following figure shows the number of valid samples collected by date. This figure was added as a supplemental figure in the supporting information.

Issue 3: Different sample sets were used in the two stages before and after. It may be due to changes in platform distribution methods and other reasons that the sample structure before and after is quite different (such as age, gender, region, etc.), resulting in a lack of comparability in the analysis. Therefore, it is recommended to explain the risk. 

Response: We agree with the reviewer that the sample structure before and after the ease of restrictions is quite different. We have added a supplemental table to shows the demographics before and after the ease of restrictions. We acknowledged the limitation of this and caution of the interpretation: “The sample sets used in the two stages before and after the easing of restrictions were quite different in demographics (see S1 Table in the supplemental information). Therefore, association between the ease of restrictions and mental health patterns cannot necessarily be considered causal relationships.” (Line 268-271, Page 25).

Issue 4: Table 3 does not give regression statistical parameters, such as R2, F value, etc.

Response: Table 3 shows results from the multinomial logistic regression to examine how individual and regional factors could predict mental disorder patterns. The R2 and F-value are measures of fit for regular regression using OLS (ordinal least square) estimation method. The logistic regression is usually estimated by maximum likelihood method. The measures of fit for logistic regression differ from the measures of fit for regular regression. There are two categories of measure of fit for logistic regression: measures of predictive power (like R2) and goodness of fit tests (like the Person Chi-square). There are many different ways to calculate R2 for logistic regression and, unfortunately, no consensus on which one is best. Mittlbock and Schemper (1996) reviewed 12 different measures [1]. Menard (2000) considered several others [2]. As for goodness of fit, the popular one is Hosmer and Lemeshow (HL) test. Hosmer and Lemeshow (1980) proposed grouping cases together according to their predicted values from the logistic regression model [3]. Specifically, the predicted values are arrayed from lowest to highest, and then separated into several groups of approximately equal size. Ten groups is the standard recommendation. HL test is shown to have some serious problems [4]. The most obvious problem is that results can depend markedly on the number of groups, and there is no theory to guide the choice of that number. 

There is rarely a fixed cut-off that distinguishes an acceptable model from one that is not acceptable. This is why that measures of fit were not as frequently being reported in logistic regression as in regular regression. For your information, the R2 for multinomial logistic regression are 2.9% and 2.1% for the sample before and after the ease of restrictions respectively. The H-L test statistics are 125.77 and 141.11 for the two sample sets respectively and both with degree of freedoms of 24.

Reference

[1] Mittlbock, M. and M. Schemper (1996) “Explained variation in logistic regression.” Statistics in Medicine 15: 1987-1997.

[2] Menard, S. (2000) “Coefficients of determination for multiple logistic regression analysis.” The American Statistician 54: 17-24.

[3] Hosmer D.W. and S. Lemeshow (1980) “A goodness-of-fit test for the multiple logistic regression model.” Communications in Statistics A10:1043-1069.

[4] Hosmer, D.W., T. Hosmer, S. Le Cessie and S. Lemeshow (1997). “A comparison of goodness-of-fit tests for the logistic regression model.” Statistics in Medicine 16: 965–980.

Reviewer 2:

1. Abstract:

• Suggest to add brief information about the mental health issues has become a major health concerns during the COVID-19 pandemic.

• Suggestion to change the last sentence in background: The findings from this research study have provided the empirical evidence …………………, and information of the factors associated with these patterns.

Response: We have revised the abstract according to your suggestion. 

2. Introduction:

• Page 11 ( page #22 in paper), Line 40 : Suggest to change “To our knowledge, no studies have been conducted to examine the …”

Response: Thanks. Done.

• Page 11 (page#22 in the paper), Line 41-43: Suggest to delete commas and change “ The objectives of the study were to provide empirical evidence of the patterns of mental health in general population both before and after easing restrictions and to examine the factors at both individual and regional levels that contributed to and/or mitigated these mental health patterns.

Response: Thanks. Done.

3. Methods:

• Page 12 (page #23 in paper), Line 58: Exclusion criteria – the authors may include the reasons why these groups of population (b. being aged < 20 years and being married, divorced or widowed; c. being aged < 20 years and having a degree of Master or PhD) were excluded from this current study.

Response: We have added justifications for these exclusion criteria: “Exclusion criteria b) and c) were set because these participants failed to pass the internal consistent checks. Those participants, who were younger than 12 years or completed questionnaires in ≤ 50 seconds, were also excluded because of data quality controls.” (Line 109-111, Page 4).

• Page 13 (page #24 in paper), Line 75: suggest to change “LCA was also used to classify individuals into distinct classes.”

Response: We have made the suggested change (Line 125, Page 5).

4. Results:

• Page 14 (page # 25 in paper), Line 116: Suggest to specify which classes. E.g., However, after easing restrictions as shown in Table 2, females had a greater chance of being in both “High Risk (Class one) and Stressed, Social Dysfunction and Unhappy (Class Two)”.

Response: Thanks. Done.

• Page 15: (page #26 in paper), Line 132: Suggest to provide additional information on the age range/group for seniors and teenagers and include them in the methods section.

Response: In our data analyses, we have not made any cutoffs of age to form age groups. The age is treated as a continuous variable in the regression analyses. Because the estimated coefficient is negative for the linear slope and positive of quadratic slope of age, the relationship between age and likelihood of being any of the mental disorder risk classes are U-shape. Therefore, we concluded that the seniors and teenagers had increased likelihood of being any of the mental disorder risk classes (Classes One, Two and Three).

• Page 17: (page# 28 in paper), Line 178: Suggest to change the word “children” to “teenagers or young adults” because people who are younger than 12 years old were excluded from the study; Several terms were used to represent a “young adults” group, such as children, youths, teenagers, as well as terms for “seniors”, such as elderly people – suggest to keep them consistent throughout this paper.

Response: As you suggested we have changed the word “children” to teenagers or young adults” in the discussion section.

5. Discussion:

• Further to the limitations of this current study, it would be helpful to include the future research studies and/or lines of work that should be considered.

Response: We have added the following future directions: “Future studies should recruit a representative probability sample or use other social media such as Weibo in order to draw more reliable conclusions”; “The longitudinal studies with follow-up assessments at different periods of pandemic are needed to determine the transition of mental health patterns and the long-term mental health outcomes.” (Line 265-275, Page 24-25)

6. Overall:

This paper is clear, well founded that provides the essential information on the mental health patterns and associated factors in a large general population both before and after easing COVID-19 restrictions. Given the potential value of the current study, I strongly recommend that the authors will take the above suggestions into consideration and revise the manuscript, in order to improve this interesting research work.

Response: Thanks.

---

## [Decision Letter · Decision Letter 1]

13 Jul 2021

Patterns of Mental Health Problems Before and After Easing COVID-19 Restrictions: Evidence from a 105248-subject Survey in General Population in China

PONE-D-21-14417R1

Dear Dr. Jiang,

We’re pleased to inform you that your manuscript has been judged scientifically suitable for publication and will be formally accepted for publication once it meets all outstanding technical requirements.

Kind regards,

Wen-Jun Tu

Academic Editor

PLOS ONE

Additional Editor Comments (optional):

Reviewers' comments:

Reviewer's Responses to Questions

**Comments to the Author**

1. If the authors have adequately addressed your comments raised in a previous round of review and you feel that this manuscript is now acceptable for publication, you may indicate that here to bypass the “Comments to the Author” section, enter your conflict of interest statement in the “Confidential to Editor” section, and submit your "Accept" recommendation.

Reviewer #1: All comments have been addressed

2. Is the manuscript technically sound, and do the data support the conclusions?

Reviewer #1: Yes

3. Has the statistical analysis been performed appropriately and rigorously? 

Reviewer #1: Yes

4. Have the authors made all data underlying the findings in their manuscript fully available?

Reviewer #1: Yes

5. Is the manuscript presented in an intelligible fashion and written in standard English?

Reviewer #1: Yes

6. Review Comments to the Author

Reviewer #1: (No Response)

7. PLOS authors have the option to publish the peer review history of their article (what does this mean?). If published, this will include your full peer review and any attached files.

Reviewer #1: **Yes: **Bo Chen

---

## [Editor Report · Acceptance letter]

21 Jul 2021

PONE-D-21-14417R1 

Patterns of mental health problems before and after easing COVID-19 restrictions: Evidence from a 105248-subject survey in general population in China 

Dear Dr. Jiang:

I'm pleased to inform you that your manuscript has been deemed suitable for publication in PLOS ONE. Congratulations! Your manuscript is now with our production department. 

Kind regards, 

on behalf of

Dr. Wen-Jun Tu 

Academic Editor

PLOS ONE